# Automatic Levothyroxine Dosing Algorithm for Patients Suffering from Hashimoto’s Thyroiditis

**DOI:** 10.3390/bioengineering10060724

**Published:** 2023-06-14

**Authors:** Ravi Sharma, Verena Theiler-Schwetz, Christian Trummer, Stefan Pilz, Markus Reichhartinger

**Affiliations:** 1Institute of Automation and Control, Graz University of Technology, 8010 Graz, Austria; markus.reichhartinger@tugraz.at; 2Division of Endocrinology and Diabetology, Department of Internal Medicine, Medical University of Graz, 8010 Graz, Austria; verena.schwetz@medunigraz.at (V.T.-S.); christian.trummer@medunigraz.at (C.T.); stefan.pilz@medunigraz.at (S.P.)

**Keywords:** Hashimoto’s thyroiditis, HPT-axis, mathematical thyroid model, discrete controller, robust stability

## Abstract

Hypothyroidism is a condition where the patient’s thyroid gland cannot produce sufficient thyroid hormones (mainly triiodothyronine and thyroxine). The primary cause of hypothyroidism is autoimmune-mediated destruction of the thyroid gland, referred to as Hashimoto’s thyroiditis. A patient’s desired thyroid hormone concentration is achieved by oral administration of thyroid hormone, usually levothyroxine. Establishing individual levothyroxine doses to achieve desired thyroid hormone concentrations requires several patient visits. Additionally, clear guidance for the dosing regimen is lacking, and significant inter-individual differences exist. This study aims to design a digital automatic dosing algorithm for patients suffering from Hashimoto’s thyroiditis. The dynamic behaviour of the relevant thyroid function is mathematically modelled. Methods of automatic control are exploited for the design of the proposed robust model-based levothyroxine dosing algorithm. Numerical simulations are performed to evaluate the mathematical model and the dosing algorithm. With the help of the developed controller thyroid hormone concentrations of patients, emulated using Thyrosim, have been regulated under the euthyroid state. The proposed concept demonstrates reliable responses amidst varying patient parameters. Our developed model provides a useful basis for the design of automatic levothyroxine dosing algorithms. The proposed robust feedback loop contributes to the first results for computer-assisted thyroid dosing algorithms.

## 1. Introduction

Hypothyroidism affects up to 5% of the general population, with an additional 5% remaining undiagnosed and is more common in women than in men [1,2]. Iodine deficiency is the most common cause of all thyroid disorders, but in iodine sufficiency, Hashimoto’s thyroiditis is the most common cause of thyroid failure, which results in hypothyroidism [2].

The thyroid gland is a butterfly-shaped gland located in front of our neck. It is responsible for generating the thyroid hormones thyroxine ($T_{4}$) and triiodothyronine ($T_{3}$) mainly. The blood serum concentration of the thyroid hormones is maintained in the human body with the help of the hypothalamic–pituitary–thyroid axis (HPT axis), see Figure 1. The concentration of $T_{3}$ and $T_{4}$ hormones acts as a negative feedback signal that feeds to the hypothalamus and pituitary gland. Corresponding to this negative feedback, the hypothalamus and the pituitary gland secrete the thyrotropin-releasing hormone (TRH) and thyroid-stimulating hormone (*TSH*), respectively. When the concentration of $T_{4}$ and $T_{3}$ increases, due to this biological closed-loop (HPT-axis), the concentration of TRH and *TSH* decreases, which results in a decrease in $T_{4}$ and $T_{3}$ hormone concentrations.

### 1.1. Medical Problem

The thyroid gland consists primarily of follicle cells. These follicle cells are responsible for the production of $T_{3}$ and $T_{4}$ hormones. When *TSH* reaches the follicle cells located in the thyroid gland, it starts stimulating follicle cells to produce $T_{3}$ and $T_{4}$ hormones. In the case of Hashimoto’s thyroiditis, the human immune system exerts autoimmunological effects on the thyroid gland including, amongst others, the production of thyroid peroxidase antibodies (TPOAb) and thyroglobulin antibodies (TgAb). These antibodies are partially responsible for blocking processes responsible for thyroid hormone production within the follicle cells [3]. This blocking of *TSH* receptors results in a decrease in $T_{3}$ and $T_{4}$ hormone concentrations (less than the concentration required by human body, i.e., $T_{4}$ and $T_{3}$, 5–12 μg/dL and 0.8–1.9 ng/mL, [4], correspondingly). Due to the HPT-axis, the hypothalamus and pituitary gland increase TRH and *TSH* secretion. Hence, the blood serum concentration of *TSH* exceeds its normal range (i.e., 0.5–5.0 mU/L see e.g., [4]). In the case of Hashimoto’s thyroiditis patients are diagnosed with low $T_{3}$ and $T_{4}$ concentrations and high *TSH* concentration. The diagnosis of Hashimoto’s thyroiditis is usually made by confirming present hypothyroidism (i.e., subclinical hypothyroidism with elevated *TSH* and decreased thyroid hormone concentrations) in conjunction with elevated TPOAb and TgAb.

Thyroid hormones play a critical role in, e.g., growth and energy metabolism [5]. Therefore, changes in the concentration of thyroid hormones can lead to multiple adverse clinical consequences such as reduced quality of life with symptoms such as fatigue, increased cardiovascular risk and weight gain [6]. A so-called Myxedema coma can be the outcome of a long-standing untreated and severe hypothyroidism [7].

In clinical routine, hypothyroid patients suffering from Hashimoto’s thyroiditis usually take levothyroxine ($LT_{4}$) oral tablets on a regular, i.e., usually daily, basis. The amount of medication ($LT_{4}$ dosage) is regulated by physicians according to the blood serum concentration of *TSH*, free thyroxine ($FT_{4}$) and free triiodothyronine ($FT_{3}$), as well as some clinical parameters such as body weight. Therefore, a visit to the physician is required under the current treatment procedure. The frequency of these visits depends on the condition of the patient’s thyroid gland (e.g., whether the patient is suffering from overt or subclinical hypothyroidism) and stages of treatment (i.e., at the initial stages, monthly visits are usually required, later, even once or twice a year is also possible).

This study aims to develop an automated dosage recommendation system that obeys all the requirements of $LT_{4}$ dosing (see Section 3.3.1). For the systematic construction of the automated dosage recommendation system, a mathematical model is developed that captures the desired human body parameters. Later, with the help of this developed mathematical model, a mathematical model-based control strategy is designed. Both the model and the control strategy are validated using the Thyrosim model [4,8,9,10] and numerical simulation framework.

The development of an automated levothyroxine dosage recommendation system will decrease patients’ dependency on in-person meetings with physicians. That will eventually lead to an improvement in the current treatment methodology, in which frequent visits to physicians are required (i.e., 4–6 weeks in the initial stages of treatment and twice a year in later stages of treatment).

### 1.2. Existing Mathematical Models of Human Thyroid Hormone Regulation

A compartment model was developed by Pandiyan [11] to describe thyroid hormone regulation in patients suffering from Hashimoto’s thyroiditis. This mathematical model does not encapsulate the effect of exogenously administered thyroxine hormone ($LT_{4}$) on patient dynamics, which is the main scope of our study. The mathematical model presented in [12] captures both the effect of $LT_{4}$ dosage on *TSH* and $FT_{4}$ dynamics. However, the *TSH* dynamics is not required in the results presented in this paper.

In [13], a mathematical model-based optimal $LT_{4}$ and levotriiodothyronine ($LT_{3}$) therapy for patients suffering from hypothyroidism was designed. The study addresses the problem of identifying the optimal dosages of $LT_{4}$ and $LT_{3}$ and analysing the effect of $LT_{4}$ monotherapy over $LT_{4}$/$LT_{3}$ combined therapy on blood serum concentrations of thyroid parameters. A model predictive controller (MPC) is employed as an optimal dosage recommender, which determines the optimal dosage of $LT_{4}$ and $LT_{3}$ concerning thyroid hormone concentrations. The model used in the MPC control algorithm is presented in [14,15,16]. In contrast to the model proposed in this paper, this MPC-employing model is much more complex and captures dynamics of human parameters which are typically measured during the regular patient’s appointments (e.g., *TSH*, $FT_{4}$). This additionally increases the computational costs of the MPC algorithm. In this paper, it is demonstrated that a control approach which is comparatively less computationally costly is still able to robustly keep the patient’s thyroid hormone concentration within the reference range.

#### Numerical Simulation of Hashimoto Patients Using Thyrosim

Thyrosim is a web application developed by UCLA laboratory. It was designed and developed primarily for researchers, clinicians and others who are interested in thyroid hormone regulation. It is a well-validated simulation of human thyroid hormone regulation [17]. Thyrosim has demonstrated its ability to reproduce a wide range of clinical data studies and can display hormonal kinetic responses to different oral dosages of $LT_{4}$ and $LT_{3}$ [17]. Thyrosim provides dynamic responses of total thyroxine ($TT_{4}$), total triiodothyronine ($TT_{3}$), $FT_{4}$, $FT_{3}$ and *TSH* over time, for a 70 kg human [9,10]. It has also been upgraded to predict $LT_{4}$ and $LT_{3}$ replacement therapy in paediatric patients [8].

Thyrosim is based on a highly complex mathematical model. This mathematical model consists of 13 differential equations and 60 different mainly patient-specific parameters [4,8,9,10]. Therefore, designing a controller on the basis of this mathematical model is a highly complicated task. This complexity in Thyrosim’s mathematical model motivated us to develop a mathematical model of human thyroid hormone regulation, which captures the main $FT_{4}$-dynamics and can be employed as a controller design model.

## 2. Methods

### 2.1. Development of an Automated Dosage Recommendation System

This study aims to develop an automated dosage recommendation system for the hypothyroid patients suffering from Hashimoto’s thyroiditis. For designing such an automated dosage recommendation system, a mathematical model of human thyroid hormone regulation has been developed. With the help of this developed mathematical model, a model-based controller was designed. This designed controller will serve as an automated dosage recommendation system. In this study, Thyrosim is employed as a so-called “virtual” patient to validate both the model and the dosing algorithm.

### 2.2. Simulation of a Patient Suffering from Hashimoto’s Thyroiditis

For simulating a patient suffering from clinical hypothyroidism, the $T_{4}$ and the $T_{3}$ secretion rates have been adjusted by using the graphical interface provided in the Thyrosim web application (http://biocyb1.cs.ucla.edu/thyrosim/cgi-bin/Thyrosim.cgi, accessed on 10 May 2022). The values of $T_{4}$ and $T_{3}$ secretion rates is 17% of the thyroid gland’s total secretion rate (which demonstrates that the patient is dependent on exogenous administration of thyroid hormones). Correspondingly, the absorption rate of oral $LT_{4}$ dosage is set at 83% (the bioavailability of oral $LT_{4}$ dosage in hypothyroid patients is slightly more than 80% [18]).

### 2.3. Validation of Developed Dosing Strategy

With the help of parameter variation in Thyrosim’s mathematical model, 50 different patients are simulated and statistical analysis is performed. All the simulated patients are treated with our developed automated dosing strategy. The developed system should be compatible with all the possible parametric uncertainties and must be able to maintain patients’ $FT_{4}$ concentration within the reference range.

## 3. Results

In this section, the development of the mathematical model-based dosing algorithm is described. In order to evaluate the functioning of this developed dosing algorithm, it is implemented with Thyrosim and simulation-based results are presented.

The results of this study were obtained using MATLAB/Simulink^®^, version 9.5.0.1298439 (R2018b) on a standard computer (Intel(R) Core(TM) i7-5600U CPU @ 2.60 GHz processor with 8 GB RAM).

### 3.1. A Control-Oriented Dynamic Model of the Thyroid Gland

A hypothyroid patient suffering from Hashimoto’s thyroiditis is dependent on the exogenous source of $T_{4}$ hormones ($LT_{4}$ tablets mainly). Therefore, the developed mathematical model needs to capture the effect of exogenous thyroxine ($LT_{4}$) on the $T_{4}$ dynamics. The well-known Michaelis-Menten equation describes enzyme-substrate relations and has been successfully used for modelling reaction dynamics on an abstract level i.e., without a too detailed description of the considered process [19]. Therefore, the Michaelis–Menten equation [19] is also used for developing this model. The developed mathematical model consists of two state variables. The total blood serum concentration of exogenous $T_{4}$ is represented by the state variable $x_{1}$, the variable $x_{2}$ captures the total $FT_{4}$ concentration in blood serum. The proposed model is given by
(1a)$$\begin{array}{c} \frac{dx_{1}}{dt}=d-k_{exc}x_{1}, \end{array}$$
(1b)$$\begin{array}{c} \frac{dx_{2}}{dt}=\frac{v_{max}TSH}{k_{m}+TSH}+k_{reac}x_{1}-k_{sec}x_{2}, \end{array}$$
where
$k_{exc}$ stands for elimination rate of orally administrated levothyroxine (h${}^{-1}$),$k_{sec}$ is the natural elimination rate of $FT_{4}$ (h${}^{-1}$),$v_{max}$ represents the maximum rate achieved by the system (μmol/h),$k_{m}$ is the Michaelis constant (μmol),$k_{reac}$ models the effect of orally administered $LT_{4}$ (ppm).

The dosage input *d* (μmol) is the exogenously infused input rate of oral $LT_{4}$. This exogenous administration of the $T_{4}$ hormone has a significant positive effect on the patient’s total thyroxine ($TT_{4}$) concentration. The $FT_{4}$ concentration in blood serum is approximately 0.02% of $TT_{4}$ concentration [20]. Therefore, an increase in $TT_{4}$ concentration will also increase the patient’s $FT_{4}$ concentration. The patient’s total $FT_{4}$ concentration is a combination of $FT_{4}$ concentration produced endogenously by the thyroid gland (represented by the Michaelis–Menten equation [19]) and the positive effect of total exogenous $T_{4}$ hormone in the blood (with the help of $LT_{4}$ tablets). For hypothyroid patients, the half-life of $T_{4}$ hormone is 9–10 days [21], which represents the natural elimination rate $k_{sec}$ of $FT_{4}$.

In the developed model, $v_{max}$, $k_{m}$ and $k_{reac}$ are patient-specific constants. The values of these parameters must be identified according to the considered patient. As already mentioned above, Thyrosim is regarded as a “virtual” patient. Therefore, the values of these patient-specific parameters are identified with the help of Thyrosim.

#### 3.1.1. Parameter Identification

Some parameters ($k_{m}$, $v_{max}$ and $k_{reac}$) of the developed mathematical model need to be identified with respect to an individual patient. In this section, a least-squares algorithm is used to minimise the cost function, see Figure 2.
(2a)$$\begin{array}{cc} J(p) & =\sum_{i=0}^{N}e_{i}^{2} \end{array}$$
with,
(2b)$$\begin{array}{cc} e_{i} & =y\left(i\right)-y_{m}\left(i\right)\;\mathrm{and}\;P={\left[\begin{array}{ccccc} k_{reac} & \; & k_{m} & \; & v_{max} \end{array}\right]}^{T} \end{array}$$
where y(i) is the output ($FT_{4}$ concentration) by Thyrosim for input dosage *d*. The signal $y_{m}\left(i\right)$ represents the output ($FT_{4}$ concentration) generated by the developed model for the same input dosage *d* and *TSH* (an output generated by Thyosim corresponding to input dosage *d*) at hour *i*. The constant *N* represents the time duration of the performed simulation.

##### Optimization Problem

For identifying the parameters $k_{reac}$, $k_{m}$ and $v_{max}$, the optimization problem
$$\begin{array}{ccc} \arg\;\min_{p} & J(p) & \\ & s.t. & \\ & & \left[\begin{array}{c} 0\\ 0\\ 0 \end{array}\right]<p\leq \left[\begin{array}{c} k_{reac_{max}}\\ k_{m_{max}}\\ v_{max_{max}} \end{array}\right] \end{array}$$
needs to be solved. The implementation of minimization of the cost function J(p) was realized with the Matlab function ‘fminsearch’. The optimization constraints need to be defined in such a manner that the value of estimated parameters must be “realistic” (e.g., $v_{max}$, $k_{m}$ and $k_{reac}$ must be positive). The identified values of parameters are listed in Table 1.

##### Model Assessment

A comparison between the $FT_{4}$ concentration generated by the proposed model and Thyrosim is presented to assess the quality of the proposed model.

In the assessment of the proposed model, we provided the same amount of dosage to Thyrosim and to the proposed model (1). In Figure 3, both mathematical models received 50 μg/day of dosage in the first month, 80 μg/day in the second month and 110 μg/day in the third month, see Figure 3a. The values of the identified parameters ($k_{m}$, $v_{max}$ and $k_{reac}$) are kept constant throughout the simulation. Thyrosim also captures daily hormonal variations of thyroid hormone parameters. Note that the proposed model in Equation (1) does not replicate these daily hormonal variations of $FT_{4}$ concentration, see Figure 3b.

From the results depicted in Figure 3, it is observable that the $FT_{4}$ concentration generated by the proposed model always lies in the range of daily hormonal variations of $FT_{4}$ concentration (generated by Thyrosim). As a result, the proposed model can be used as a mathematical model for designing an automated dosage recommendation system for Thyrosim, which prescribes a daily dosage.

### 3.2. Some Basic Control System-Related Properties of the Model

Analysing control theory-related system properties of the proposed mathematical model is an intended part of controller design. The system’s eigenvalues characterize its response to a given input and non-vanishing initial settings of the state variables, whereas controllability describes whether the system can be guided from an arbitrary initial state to a target state over a finite time interval using the actuating signal [22]. The system should be controllable to attain the desired final state by using a controlled input. Note that the proposed model can be rewritten as
(3a)$$\begin{array}{cc} \dot{x} & =Ax+bd+Vp \end{array}$$
(3b)$$\begin{array}{cc} y & =c^{T}x \end{array}$$
with,
$$A=\left[\begin{array}{ccc} -k_{exc} & & 0\\ k_{reac} & & -k_{sec} \end{array}\right]\;\;,\;b=\left[\begin{array}{c} 1\\ 0 \end{array}\right]$$
$$V=\left[\begin{array}{c} 0\\ v_{max} \end{array}\right]\;\;,\;c^{T}=\left[\begin{array}{ccc} 0 & & 1 \end{array}\right]$$
$$\mathrm{and}p=\frac{TSH}{km+TSH}$$
Some system properties are:(a)The eigenvalues of the system are $-k_{exc}$ and $-k_{sec}$, which represent how fast the system will react in response to provided $LT_{4}$ dosage;(b)Controllability: It can be easily checked that the developed system is controllable. Therefore, it is possible to achieve a certain level of $FT_{4}$ concentration in a finite time interval by using a controlled $LT_{4}$ dosage.

### 3.3. Controller Design

This study aims to develop an automated dosage recommendation system that enables us to automate the dosage of $LT_{4}$. Exogenous administration of hormones is a highly sophisticated task. The developed controller must follow all the desired guidelines mentioned in the FPI (full prescription information) of $LT_{4}$ dosing (e.g., the dosage must be recommended incrementally, the sampling time must be according to patient condition). Therefore, the designed controller must obey all the following desired requirements:

#### 3.3.1. Requirements for an Automated Dosing Strategy

(i)The mathematical model (1) consists of different biological parameters that vary from patient to patient. Therefore, a “robust” dosage administration is mandatory;(ii)Oral administration of hormones will result in a sudden increment of patient blood serum concentration of $FT_{4}$. Therefore, the designed controller is not allowed to react “too fast”;(iii)A larger daily dosage exceeding 150 μg in children and 500 μg in adults may produce serious or even life-threatening manifestations of toxicity when administered for longer time periods [23]. Consequently, limitations on the prescribed dosage are essential. Initially, at the beginning of the therapy, a small amount of dosage is prescribed. These dosages are increased incrementally (12.5 μg to 25 μg) until the patient reaches the euthyroid state. Therefore, a defined rate limiter is necessary in the design of an automated dosage recommendation system to control the change in the recommended dosage. Dosages are recommended to the patient with the help of blood serum concentrations of different thyroid hormone parameters ($FT_{4}$, TPOAb, *TSH*, etc.). Measurement of these parameters initially take place every 4–6 weeks (this sampling time increases in the later stages of treatment, once or twice a year is also possible). Once the parameters are measured, the physician recommends an amount of $LT_{4}$ dosage. This recommended dosage must be constant until the next measurement of thyroid parameters. Therefore, it is required to design a discrete-time controller with a sampling rate of between 4 and 6 weeks;(iv)A dosage above 200 μg per day must require consultation with a physician before recommendation. Therefore, a saturator is an intended part of this proposed dosage recommendation system that saturates dosage above 200 μg per day;(v)Steady-state requirements: When the patient reaches euthyroid state (normal condition of thyroid hormones), the amount of the dosage recommended to the patient should be constant, unless there are some variations in blood serum concentrations of thyroid hormones.

#### 3.3.2. Selection of the Controller Type

The recommended dosage must be constant between the two diagnoses of patient thyroid parameters (measurement of thyroid gland parameters). Only after the measurement of the thyroid hormones is a change in the recommended amount of dosage possible. Therefore, designing a discrete-time controller is a necessary part of this study. The discrete-time controller’s sampling time must correspond to the patient’s laboratory measurement interval (initially, the diagnosis time is 4–6 weeks, it will increase in the later stages of treatment).

The goal behind designing an automated dosage recommendation is not to achieve a particular $FT_{4}$ concentration but to achieve the euthyroid state (9.2 to 16.0 ng/L [24], depending upon patient’s body mass index (BMI) and other parameters such as age or history of chronic disease). Hence, it is recommendable to design a non-integrating discrete-time controller with a sampling time of 30 days (initially, monthly visits to the physician are required).

##### Calculating the Continuous-Time Transfer Function of the Proposed Model

The proposed model is a so-called linear MISO system (Multi-Input and Single Output). From Equations (3a) and (3b), the output (the Laplace transform of the function y(t) is denoted by $\bar{y}\left(s\right)$, i.e., $\bar{y}\left(s\right)$=L{y(t)}) y(t) in the Laplacian domain can be computed by
(4a)$$\bar{y}\left(s\right)=\left[\begin{array}{ccc} G(s) & \; & S(s) \end{array}\right]\left[\begin{array}{c} \bar{d}\left(s\right)\\ \bar{p}\left(s\right) \end{array}\right]$$
with the transfer functions
(4b)$$\begin{array}{cc} G(s) & =\frac{k_{reac}}{s^{2}+\left(k_{sec}+k_{exc}\right)s+k_{sec}k_{exc}}\;\mathrm{and}\;S\left(s\right)=\frac{v_{max}}{s+k_{sec}}. \end{array}$$
The transfer function G(s) is regarded as the plant transfer function and S(s) represents the transfer function capturing the dynamics from *TSH* to the output *y*.

If the patient is suffering from clinical hypothyroidism, most of the $FT_{4}$ concentration presented in the patient body will be due to the recommended dosage (exogenous infusion of thyroxine). Hence, we assume the effect of *TSH* disturbance (endogenously produced $FT_{4}$) to be negligible.

##### Discretization of Plant

The patient is allowed to take one tablet of $LT_{4}$ per day. Hence, we use the impulse-invariant discretization [25] method for discretizing the plant G(s), which results in the discrete-time transfer function.
(5a)$$\begin{array}{cc} G(z) & =\frac{A(q_{2}-q_{1})z}{z^{2}-\left(q_{1}+q_{2}\right)z+q_{1}\cdot q_{2}} \end{array}$$
with
(5b)$$\begin{array}{cc} A & =\frac{k_{reac}}{k_{sec}-k_{exc}}\;,\;q_{1}=e^{-k_{sec}\cdot T_{d}}\;,\;q_{2}=e^{-k_{exc}\cdot T_{d}} \end{array}$$
Due to the high sampling time (≈28 days), the value of $q_{1}\cdot q_{2}$ is approximately zero, which yields
(5c)$$\begin{array}{cc} G(z) & =\frac{A(q_{2}-q_{1})}{z-(q_{1}+q_{2})}. \end{array}$$
Using, $q_{2}-q_{1}=p_{1}$ and $q_{1}+q_{2}=p_{2}$ yields
(5d)$$\begin{array}{cc} G(z) & =\frac{A\cdot p_{1}}{z-p_{2}}. \end{array}$$

##### Designing the Rate Limiter

The patient’s hormone concentration is maintained with the help of a developed dosage recommendation system. Any sudden change in these hormones (e.g., $T_{4}$, *TSH* etc.) is unaffordable for the patient’s biological condition. The recommended dosage of $LT_{4}$ must be adjusted according to the mentioned requirements (see Section 3.3.1). Hence, the development of a designated rate limiter is an intended part of this research. Using the unity feedback loop as depicted in Figure 4 with the transfer function,
$$K_{r}\left(z\right)=\frac{k_{r}}{z-1}$$
the rate limitation is realized within the automated dosing algorithm. The parameter $k_{r}$ is the maximum admissible change in the output (e.g., recommended dosage). Once $k_{r}$ is defined, the change in output is ensured to not exceed the defined value of $k_{r}$ (the desired change in recommended dosage 12.5 μg to 25 μg).

The overall transfer function of the rate limiter describing the dynamics from the controller’s output to the plant’s input is given by
$$K_{rz}\left(z\right)=\frac{k_{r}}{z-\Delta }\;\;,\;\Delta =1-k_{r}.$$
In order to design the controller transfer function R(z), the rate limiter $K_{rz}\left(z\right)$ and the plant G(z) are considered as controller design transfer function (total plant in Figure 5).
(6a)$$\begin{array}{c} G_{p}\left(z\right)=K_{zr}\cdot G\left(z\right) \end{array}$$
The values of *A*, $p_{1}$ and $p_{2}$ are mentioned in Equation (5a) and
(6b)$$\begin{array}{c} G_{p}\left(z\right)=\frac{A\cdot p_{1}\cdot k_{r}}{z^{2}-\left(p_{2}+\Delta \right)z+p_{2}\cdot \Delta }=\frac{\mu_{0}}{z^{2}+v_{1}z+v_{0}} \end{array}$$
(6c)$$\begin{array}{c} \frac{\mu (z)}{v(z)}=\frac{\mu_{1}z+\mu_{0}}{z^{2}+v_{1}z+v_{0}} \end{array}$$
The pole assignment method, see, e.g., [26], is used for designing a discrete-time controller
(7a)$$\begin{array}{c} R\left(z\right)=\frac{b_{1}z+b_{0}}{a_{1}z+a_{0}}=\frac{b(z)}{a(z)}\;. \end{array}$$
The closed-loop transfer function of the developed system depicted in Figure 5 is
(7b)$$\begin{array}{c} T\left(z\right)=\frac{b(z)\mu (z)}{v(z)a(z)+b(z)\mu (z)}=\frac{\mu_{T}\left(z\right)}{v_{T}\left(z\right)}\;. \end{array}$$
Denoting $v_{T}\left(z\right)$ as the is desired characteristic polynomial of T(z), i.e.,
(7c)$$\begin{array}{c} v_{T}\left(z\right)=w_{3}z^{3}+w_{2}z^{2}+w_{1}z+w_{0}\;, \end{array}$$
the poles of the characteristic polynomial can be placed at desired locations specified by w(z). By solving
(7d)$$\left[\begin{array}{ccccccc} v_{0} & \; & 0 & \; & \mu_{0} & \; & 0\\ v_{1} & \; & v_{0} & \; & \mu_{1} & \; & \mu_{0}\\ v_{2} & \; & v_{1} & \; & 0 & \; & \mu_{1}\\ 0 & \; & v_{2} & \; & 0 & \; & 0 \end{array}\right]\left[\begin{array}{c} a_{0}\\ a_{1}\\ b_{0}\\ b_{1} \end{array}\right]=\left[\begin{array}{c} w_{0}\\ w_{1}\\ w_{2}\\ w_{3} \end{array}\right]$$
the values $a_{1}$, $a_{0}$, $b_{1}$ and $b_{0}$ and hence the controller R(z) are calculated

When the poles of characteristic polynomial $v_{T}$(z) are placed near z=0, then the system will react too fast. That is not recommendable for biological systems. When the poles of $v_{T}\left(z\right)$ are placed near −1 or +1, then the developed system has overshoots and it also negatively effects the robustness of the system, which is undesirable. Currently, all the poles of $v_{T}\left(z\right)$ are placed at z=−0.01, because at this point, the developed system shows less overshoot and the system is not reacting too fast.

#### 3.3.3. Robustness Analysis of the System

The developed controller must be suitable for all patients and must be able to deal with the possible uncertainties in the parameters of the proposed model (1). In the proposed model, the bioavailability of $LT_{4}$ drug is uncertain, which will vary from slightly above 80% to 64% [18] (hypothyroidism slightly above 80% and the under-fastened condition bioavailability of $LT_{4}$ decrease to 64%).

To analyse the robust stability of the developed system for the mentioned uncertainty (bioavailability of $LT_{4}$) the Jury, Pavlidis theorem [27] is used. This theorem states that the polynomial family p(z,Q)={p(z,q)|q∈Q} with ($p\left(z,q\right)=a_{0}\left(q\right)+a_{1}\left(q\right)z+a_{2}\left(q\right)z^{2}\ldots +a_{n}\left(q\right)z^{n}$) p(z,q) continuous is Schur-stable if and only if:(i)There exists a $q^{0}$∈Q for which the polynomial $p^{0}\left(z\right)=p\left(z,q^{0}\right)$ is Schur-stable;(ii)p(1,q)≠0, ∀ q∈Q;(iii)p(−1,q)≠0, ∀ q∈Q;(iv)det S(q)≠0, ∀ q∈Q, where S(q)=X(q)−Y(q) and (omitting the dependency on *q*)
$$X=\left[\begin{array}{ccccccccc} a_{n} & & a_{n-1} & & a_{n-2} & & .. & & a_{2}\\ 0 & & a_{n} & & a_{n-1} & & .. & & a_{3}\\ 0 & & 0 & & a_{n} & & .. & & a_{4}\\ : & & : & & : & & .. & & :\\ 0 & & 0 & & 0 & & .. & & a_{n} \end{array}\right]\;\;,Y=\left[\begin{array}{ccccccccc} 0 & & 0 & & 0 & & .. & & a_{0}\\ : & & : & & : & & .. & & :\\ 0 & & 0 & & a_{0} & & .. & & a_{n-4}\\ 0 & & a_{0} & & a_{1} & & .. & & a_{n-3}\\ a_{0} & & a_{1} & & a_{2} & & .. & & a_{n-2} \end{array}\right]$$
where matrix X(q) and Y(q) have (n−1) rows and (n−1) columns, *n* is the order of polynomial p(z,q).

The characteristics polynomial v(z)a(z)+u(z)a(z) of the developed system is
$$a_{1}z^{3}+(a_{0}-a_{1}\left(p_{2}+\Delta \right)z^{2}+\left[a_{1}p_{2}\Delta -a_{0}\left(p_{2}+\Delta \right)+Ap_{1}k_{r}b_{1}\right]z+a_{0}p_{2}\Delta +b_{0}Ap_{1}k_{r}\;.$$
The values of patient-based parameters *A*, $p_{{}^{1}}$ and $p_{2}$ are mentioned in Equations (5a) and (5d). The characteristic polynomial family $\left\{v_{T}\left(z,k_{exc}\right)|k_{exc}\in \left[0.16,0.36\right]\right\}$ is Schur-stable, if it follows all the mentioned conditions by Jury and Pavlidis [27], i.e.:(i)$v_{T}\left(z,0.26\right)$ is Schur-stable;(ii)$v_{T}\left(1,k_{exc}\right)$≠ 0, ∀ $k_{exc}$∈ [0.16, 0.36], see Figure 6a;(iii)$v_{T}\left(-1,k_{exc}\right)$≠ 0, ∀ $k_{exc}$∈ [0.16, 0.36], see Figure 6b;(iv)det S($k_{exc}$) ≠ 0, ∀ $k_{exc}$∈ [0.16, 0.36], where S($k_{exc}$) = X($k_{exc}$) − Y($k_{exc}$), see Figure 6c.

The characteristic polynomial $v_{T}\left(z\right)$ obeys all the conditions mention by the Jury, Pavlidis theorem, for $k_{exc}$∈ [0.16, 0.36]. This is also illustrated by Figure 6. Therefore, the system is robust and stable for the provided parameter uncertainty $k_{exc}$∈ [0.16, 0.36].

### 3.4. Evaluation of Automated Dosage Recommendation System

The developed automated dosage recommendation system is used to maintain the $FT_{4}$ concentration of a patient modelled by Thyrosim, see Figure 7.

Figure 8a shows the amount of dosage prescribed by the developed automated dosage recommendation system. Figure 8b represents the $FT_{4}$ concentration generated by Thyrosim for the corresponding amount of dosage. The total time of the simulation is 12 months. For the entire simulation, the reference $FT_{4}$ concentration is constant at 13 ng/L. The $LT_{4}$ dosage recommended to the patient is varied by 12.5 μg/month. The recommended amount of dosage becomes constant once the patient’s $FT_{4}$ concentration settles within the reference range.

In Figure 9a, the amount of dosage prescribed by the developed automated dosage recommendation system is presented. Figure 9b shows the $FT_{4}$ concentration generated by Thyrosim for the corresponding amount of dosage. The total time of simulation is three years. The reference $FT_{4}$ concentration is 16 ng/L for the first 18 months and in the last 18 months, it decreases to 12 ng/L. The results from Figure 8 and Figure 9 demonstrate that with the help of the developed automated dosage recommendation system, the patient’s $FT_{4}$ concentration is maintained within the reference range. For maintaining the patient’s $FT_{4}$ concentration within the reference range, the dosage recommended by the developed automated recommendation system follows all the requirements of $LT_{4}$ dosing mentioned in the Section 3.3.1.

#### Statistical Analysis of 50 Patients

In Figure 10, the parameters of the Thyrosim model are varied to simulate 50 different virtual patients. The parameters have varied within their possible range of variation. The designed automated dosage recommendation system has been used to treat all simulated patients. The mean and standard deviation of the patients’ $FT_{4}$ concentration and recommended dosage has been calculated monthly. The results reveal that while the variation between the patients’ $FT_{4}$ concentrations is high at the beginning of treatment, by employing the developed automated dosage recommendation system in the latter stages of treatment, all patients acquire the $FT_{4}$ concentration within the reference range. This demonstrates that with the possible parameter uncertainties, a certain level of $FT_{4}$ concentration (within the euthyroid range 9.2–16 ng/L [24]) can be maintained by using the developed automated dosage recommendation system.

The developed automated $LT_{4}$ dosing algorithm sampled with a fixed sampling time (e.g., 28 days). In real-world scenarios, physicians may choose to adjust the recommended amount of $LT_{4}$ dosage for patients at different time intervals (e.g., 4–6 weeks in the initial phase of the treatment, and in the later stages, even twice a year is also possible). Furthermore, the proposed mathematical model (1) does not account for the effects of thyroid antibody concentrations (TPOAb and TgAb) on the system’s dynamics. To encapsulate the effects of these antibody concentrations (TPOAb and TgAb) on the dynamics of the system, further investigation is required.

## 4. Discussion

The scope of this study is to develop an automated dosage recommendation system that prescribes a daily dosage of $LT_{4}$ to hypothyroid patients. With the help of the prescribed dosage, patients can maintain their thyroid hormone concentration within the euthyroid state. The recommended dosage must follow all the requirements of $LT_{4}$ dosing (see Section 3.3.1). In our model, the reference range of $FT_{4}$ concentration is 9.2–16 ng/L (mean 12.9 ng/L [24]), the developed dosage recommendation system must be able to maintain the patient’s $FT_{4}$ concentration within the reference range.

In Figure 8, the reference concentration is set at 13 ng/L. The proposed controller is a non-integrating discrete-time controller. As a result, the system’s output will have a steady-state error. However, the goal is not to reach a specific reference concentration; rather, a $FT_{4}$ concentration within the reference range is suitable for the patient’s biological condition. Figure 8a represents the dosage recommended by the controller. The recommended dosage follows the mentioned requirements (see Section 3.3.1). Initially, a small amount of the dosage (e.g., 12.5 μg) is recommended by the developed controller. Later, the recommended dosage is incremented with monthly increments of 12.5 μg. When the $FT_{4}$ concentration of Thyrosim attains a value within the reference range, the recommended dosage becomes constant (as mentioned in FPI) with no fluctuations. Figure 8b represents the $FT_{4}$ concentration of Thyrosim for the corresponding amount of dosage. With the help of the recommended dosage generated by the developed controller, the $FT_{4}$ concentration of Thyrosim is increased slowly after reaching the desired concentration (within the reference range of $FT_{4}$ concentration) Thyrosim’s $FT_{4}$ concentration becomes constant (as demanded in Section 3.3.1).

In Figure 9, the reference $FT_{4}$ concentration varies over time (e.g., first 18 months 16 ng/L, last 18 months 12 ng/L). Figure 9a represents the dosage recommended by the controller. In the first 18 months, a constant increment of 12.5 μg/month in dosage is observable. When Thyrosim $FT_{4}$ concentration reaches a value within the reference $FT_{4}$ concentration, the dosage recommended by the controller gets constant. In the last 18 months, the reference $FT_{4}$ concentration is decreased to 12 ng/L. Consequently, the dosage recommended by the developed controller decreases with a decrement of 12.5 μg/month. When Thyrosim’s $FT_{4}$ concentration attains a value within the reference $FT_{4}$ concentration, the recommended dosage by the controller becomes constant. This demonstrates that the system fulfils the steady-state requirement (recommended dosage must be constant after reaching the desired $FT_{4}$ concentration “euthyroid state”). Correspondingly, Figure 9b the $FT_{4}$ concentration generated by Thyrosim varies according to the recommended dosage by the controller. When Thyrosim $FT_{4}$ concentration attains a value within the reference $FT_{4}$ concentration (for both times, initially, 16 ng/L and later 12 ng/L), it becomes constant with no fluctuations. This demonstrates that the developed automated dosage recommendation system adheres to the $LT_{4}$ dosage administration guidelines (see Section 3.3.1). Thyrosim’s $FT_{4}$ concentration can be maintained within the reference range by employing this proposed automated dosage recommendation system.

The biological parameters of patients are different from patient to patient. The proposed automated dosage recommendation system must be able to deal with possible parametric uncertainties and recommend the dosage that obeys all $LT_{4}$ prescription guidelines (see Section 3.3.1). In Figure 10, a statistical analysis is performed to assess the developed system’s capability in dealing with possible parameter uncertainties. The Thyrosim model has 60 biological parameters that can be varied within their possible range of variation. This parameter variation in the Thyrosim model is used to simulate 50 different patients. In Figure 10a, at the beginning of the treatment, the difference between patients’ $FT_{4}$ concentration is high, as the biological conditions of patients can be different. As treatment progresses, the standard deviation between patients’ $FT_{4}$ concentration decreases, and in the final phases of treatment, the standard deviation between patients’ $FT_{4}$ concentration has been minimized with the help of the dosage recommended by the developed automated dosage recommendation system. Correspondingly, in Figure 10b, the standard deviation in recommended dosage is low at the beginning of the treatment. Patients with different biological conditions require different amounts of dosage to achieve reference $FT_{4}$ concentration, therefore, the standard deviation in the recommended dosage increases as treatment progresses. Thus, a reference $FT_{4}$ concentration is maintained in different patients (simulated by Thyrosim) using the developed automated dosage recommendation system.

Note that the results presented in this study are based on simulation only. Consequently, in a next step, data from real patients need to be incorporated and a clinical validation of the proposed approach is required.

## 5. Conclusions

In this study, an automated dosage recommendation system has been developed that provides $LT_{4}$ dosage to hypothyroid patients suffering from Hashimoto’s thyroiditis. The simulation-based results reveal that the developed system can maintain patients’ $FT_{4}$ concentration (emulated by Thyrosim) within the reference range. The statistical analysis of 50 different patients illustrates that despite variation in biological parameters, the proposed system can maintain patients’ $FT_{4}$ concentration within the reference range. The findings of this study provide a good basis for the development of an automated dosage recommendation system that could minimize the necessity for in-person interactions with medical professionals.

## Figures and Tables

**Figure 1 bioengineering-10-00724-f001:**
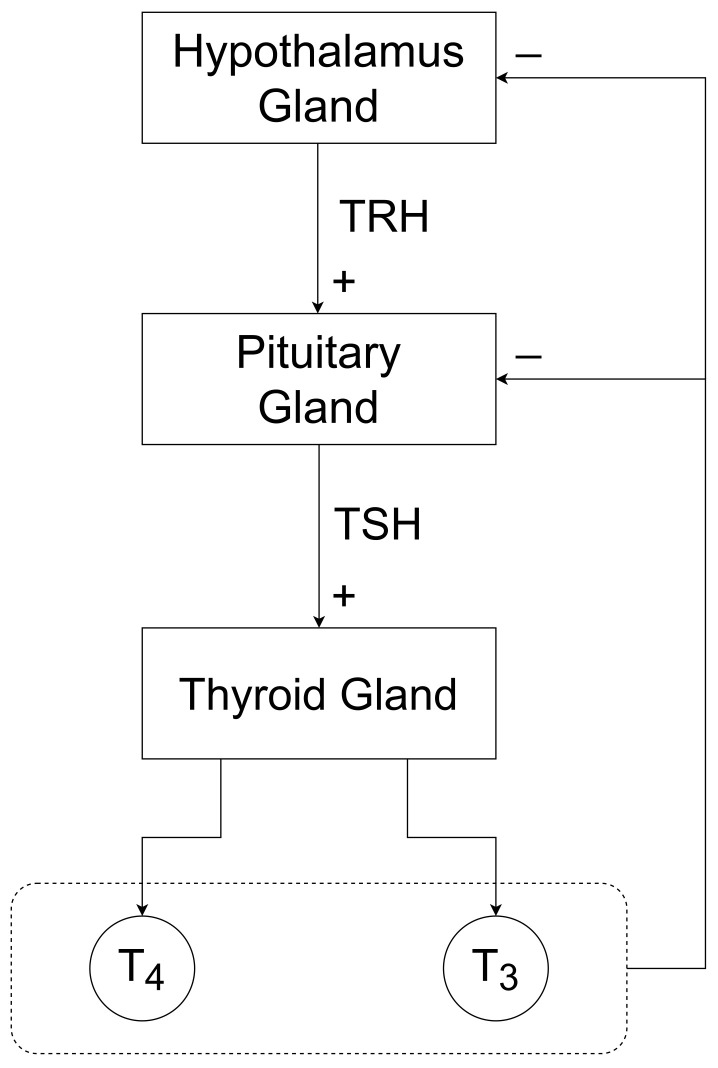
Block diagram of hypothalamic–pituitary–thyroid axis (HPT axis), explaining thyroid hormone regulation in the human body.

**Figure 2 bioengineering-10-00724-f002:**
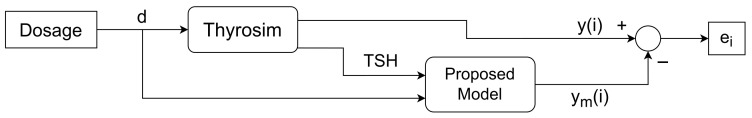
For parameter identification, both models receive the same amount of $LT_{4}$ dosage.

**Figure 3 bioengineering-10-00724-f003:**
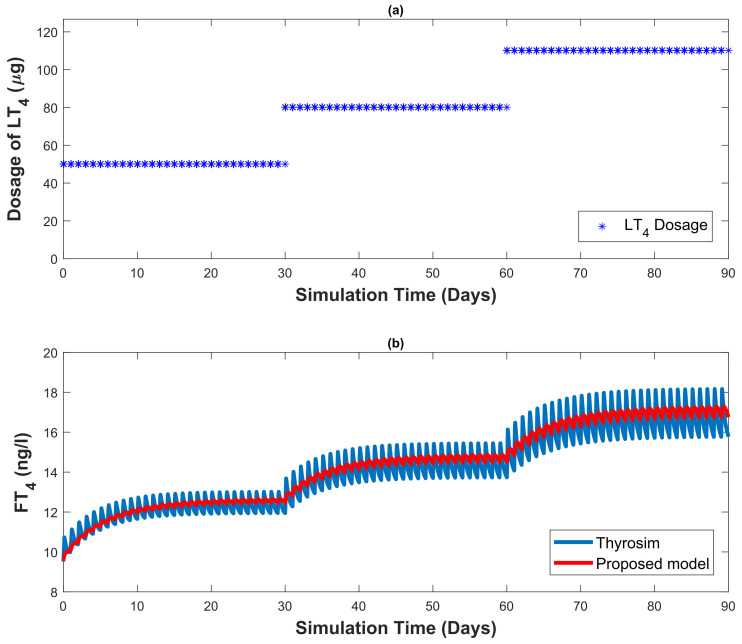
Numerical simulation is used for parameter identification. The same amount of dosage is provided to the proposed model and Thyrosim. A comparison between the two generated outputs ($FT_{4}$ concentration generated by Thyrosim and proposed model) is represented.

**Figure 4 bioengineering-10-00724-f004:**
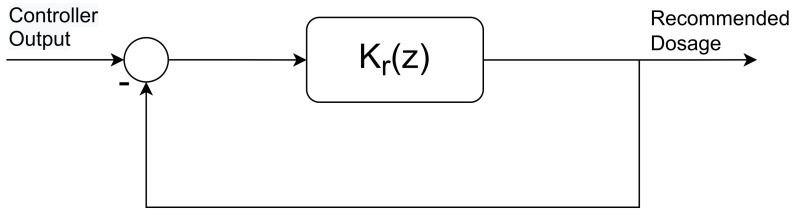
Sketch of the proposed rate limiter to control the rate of change in recommended dosage.

**Figure 5 bioengineering-10-00724-f005:**
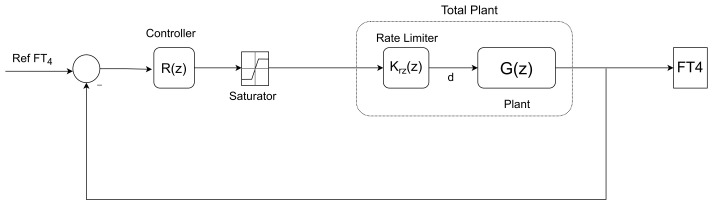
Block diagram of the designed feedback loop including saturator and rate limiter.

**Figure 6 bioengineering-10-00724-f006:**
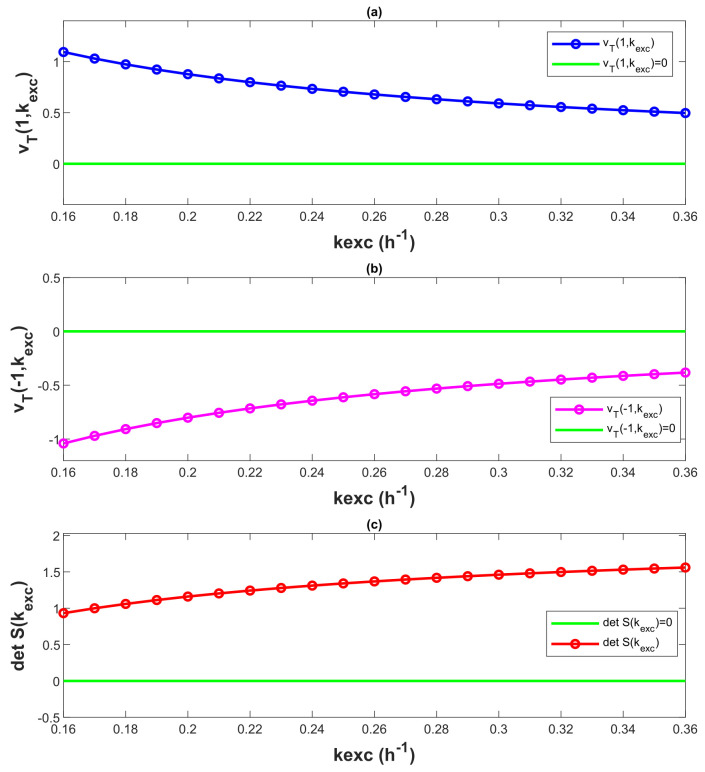
Robust stability analysis of system with Jury, Pavlidis theorem.

**Figure 7 bioengineering-10-00724-f007:**
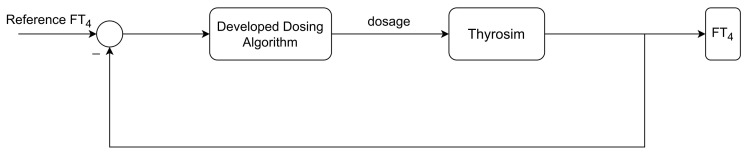
Structure of the implemented simulation setup for evaluating the designed dosing algorithm.

**Figure 8 bioengineering-10-00724-f008:**
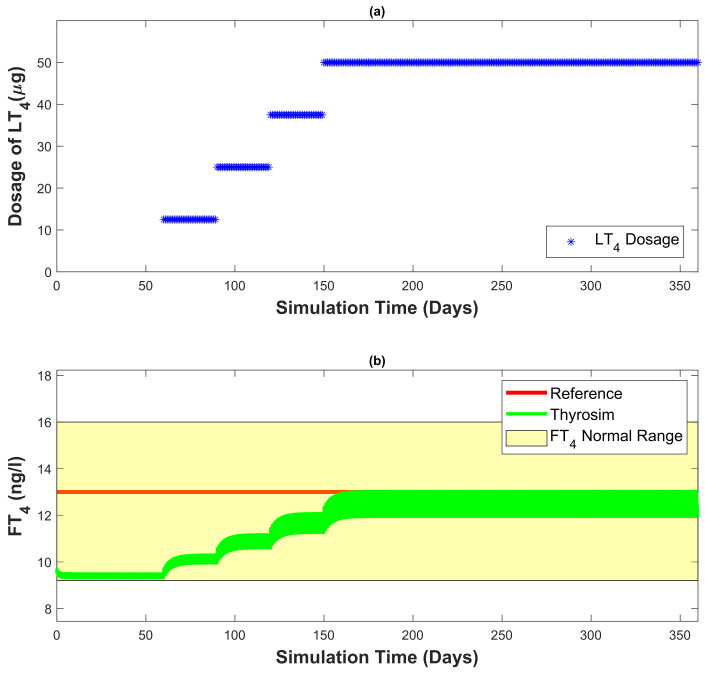
Thyrosim $FT_{4}$ concentration is regulated by the developed automated dosage recommendation system (reference $FT_{4}$ concentration 13 ng/L), simulation time of 12 months.

**Figure 9 bioengineering-10-00724-f009:**
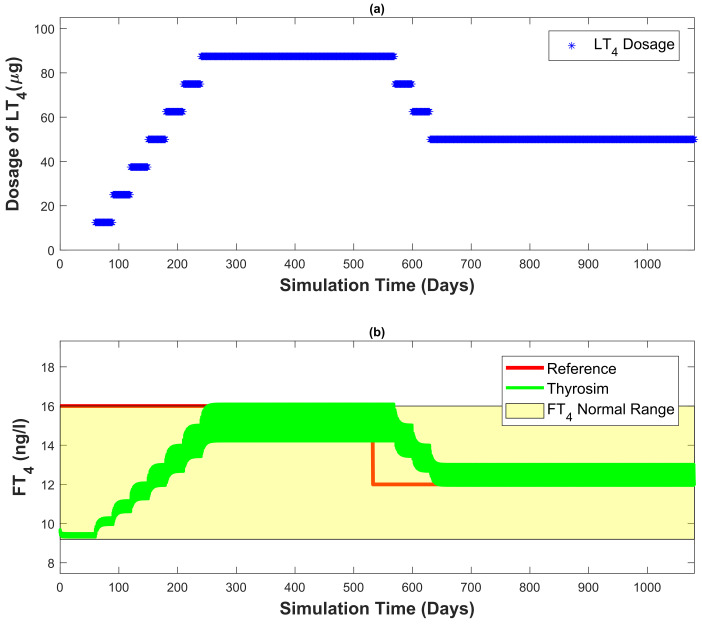
The $FT_{4}$ concentration of the patient simulated by the Thyrosim is regulated by the developed automated dosage recommendation system (reference $FT_{4}$ concentration 16 ng/L for the first 18 months and for the last 18 months, the reference $FT_{4}$ concentration is 12 ng/L).

**Figure 10 bioengineering-10-00724-f010:**
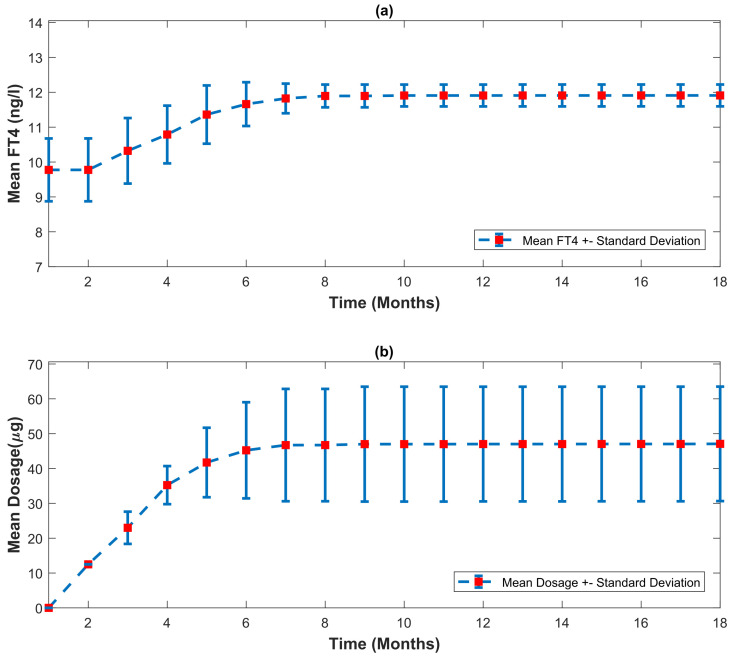
With the help of parameter variation in Thyrosim’s model, 50 patients are simulated. The developed automated dosage recommendation system is used to treat the patients. The mean and standard deviation of the patients’ $FT_{4}$ levels, as well as the recommended dosage, are presented.

**Table 1 bioengineering-10-00724-t001:** Identified parameters of the proposed model (1a), (1b) for capturing the dynamics of a patient emulated by Thyrosim.

Constants	Estimation
$k_{reac}$	7.25 ppm
$k_{m}$	9.504·$10^{3}$ μmol
$v_{max}$	2.19·$10^{-4}$ μmol/h

## Data Availability

The data presented in this study are available on request from the corresponding author.

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
