# Peer review of "Automatic Levothyroxine Dosing Algorithm for Patients Suffering from Hashimoto’s Thyroiditis"

_bioengineering, 2023, doi:10.3390/bioengineering10060724_

Round 1

Reviewer 1 Report

The paper deals with the design of an optimal control law to regulate the dose of Levothyroxine to patients suffering from Hashimoto’s Thyroiditis.

 Levothyroxine release is naturally controlled by the organism  through a negative feedback closed loop physiological control system; when such a system does not work properly, it must be supported by an artificial external help.

The authors propose a simple control-oriented model of the second order, whose parameters are optimized to reflect the properties of the more complex models available in the literature; in particular the web application Thyrosim, developed at the UCLA laboratory, is exploited to identify the model parameters. Then, a discrete-time second order controller is designed in the Z-domain  to regulate the dose of levothyroxine administered to the patient; to this end, an optimal pole placement tecnique is exploited by the authors. The proposed methodology is tested on fifty virtual patients through simulation, and showed a good performance.

The paper il clearly written, and can represent a useful advance in the field of the control of physiological systems. Obviously, a stronger validation involving real patients, is necessary in order to assess the proposed approach; the current paper can be considered a good preliminary work in this direction.

A minor point: some secions, as for example "Medical problems" at p. 2, "Numerical Simulation of Hashimoto Patients using Thyrosim" at p. 3, etc., should be numbered.

--

Reviewer 2 Report

This is an interesting study that establishes a complex mathematical model designed to facilitate the calculation of the appropriate dose of levothyroxine in patients with hypothyroidism due to Hashimoto's Thyroiditis. The authors performed multiple numerical simulations  to evaluate the mathematical model and the dosing algorithm using different patient parameters. Unfortunately, the authors do not list the limitations of their study and, of course, a major limitation is the lack of real-world testing of the method.

Mathematical models for calculating the adequate dose of levothyroxine have been proposed for some time, as the authors themselves describe, but, in practice, what will be their use? The model needs to be compared with daily practice in a cohort of patients.

I suggest that the authors discuss the limitations of their study and mention the need for validation of the model in the real world.
